# Postmortem Muscle Protein Changes as a Tool for Monitoring Sahraoui Dromedary Meat Quality Characteristics

**DOI:** 10.3390/foods11050732

**Published:** 2022-03-02

**Authors:** Hanane Smili, Samira Becila, Antonella della Malva, Ayad Redjeb, Marzia Albenzio, Agostino Sevi, Antonella Santillo, Baaissa Babelhadj, Abdelkader Adamou, Abdelghani Boudjellal, Rosaria Marino

**Affiliations:** 1Equipe Maquav, Laboratoire Bioqual, Institut de la Nutrition de l’Alimentation et des Technologies Agro-Alimentaires (INATAA), Université Frères Mentouri Constantine 1, Route Ain El-Bey, Constantine 25000, Algeria; hanane.smili@umc.edu.dz (H.S.); samira.becila@umc.edu.dz (S.B.); aboudjellal@yahoo.fr (A.B.); 2Faculté des Sciences de la Nature et de la Vie, Université Kasdi Merbah Ouargla, Ouargla 30000, Algeria; redjeb.ayad@gmail.com (A.R.); babelhadjbaaissa@gmail.com (B.B.); adamoudz@yahoo.fr (A.A.); 3Department of Agriculture, Food, Natural Resources and Engineering (DAFNE), University of Foggia, Via Napoli 25, 71121 Foggia, Italy; marzia.albenzio@unifg.it (M.A.); agostino.sevi@unifg.it (A.S.); antonella.santillo@unifg.it (A.S.); rosaria.marino@unifg.it (R.M.)

**Keywords:** Sahraoui dromedary, slaughter age, postmortem time, meat quality, myofibrillar protein changes

## Abstract

The effects of slaughter age (2 vs. 9 years) and postmortem time (6, 8, 10, 12, 24, 48 and 72 h) on the meat quality and protein changes of the *longissimus lumborum* muscles of the Algerian Sahraoui dromedary were investigated. Muscles of young dromedaries evidenced a slower acidification process and a significantly higher myofibrillar fragmentation index throughout the postmortem time. The SDS-PAGE of sarcoplasmic and myofibrillar proteins revealed that meat from young dromedaries was characterized by the lowest percentage of myoglobin (*p* < 0.001) and the highest percentage of desmin (*p* < 0.01). During postmortem time, a decrease was found for phosphoglucomutase (*p* < 0.01), α-actinin (*p* < 0.05) and desmin (*p* < 0.01) in meat from young dromedaries. Western blot revealed an intense degradation of troponin T in younger dromedaries, with an earlier appearance of the 28 kDa polypeptide highlighting differences in the proteolytic potential between dromedaries of different ages. Principal component analysis showed that meat from young dromedaries, starting from 24 h postmortem, was located in a zone of the plot characterized by higher levels of the myofibrillar fragmentation index, 30 kDa polypeptide and enolase, overall confirming greater proteolysis in younger animals. Data suggest that the investigation of the muscle proteome is necessary to set targeted interventions to improve the aging process of dromedary meat cuts.

## 1. Introduction

The camel species present in Algeria is the dromedary (*Camelus dromedarius*); its global population was estimated to be 416,519 in 2019, thus placing Algeria in the 14th rank worldwide. During the last decade, a growing demand for dromedary meat products has been observed, from 4500 tons in 2009 to 6514 tons in 2019 [1], particularly in harsh arid and semiarid areas where climate negatively impacts the production efficiency of other species. Currently, this production relates to a particular area, due to the natural distribution of the species in addition to socioeconomic aspects. The Sahraoui dromedary population has been relevant for the local economic sustainability of the northern Sahara region of Algeria; the animal is characterized as being rustic, robust and chunky, with high musculature that fits better to drought, transportation and meat production [2]. Dromedary meat is an interesting source of protein that should be exploited at its best, because it yields a heavy carcass under inexpensive management systems [3,4]. However, at present consumers perceive dromedary meat as a “low-quality product” characterized by low tenderness, because it is generally obtained from animals slaughtered at mature ages (>4 years old) and aged for very short time (24 h). A previous study on camel meat [5] revealed that the differences found due to animals’ ages could be related to histological changes that take place in muscle structures as animals mature, especially in terms of the amount of connective tissue and its deposition.

Several studies [6,7,8] have reported that tenderness is a complex attribute influenced by different factors, related on the one hand to an animal’s species, age, breed, muscle type and diet, but especially to postmortem factors (storage time, cooling rate and temperature). However, it is well-established that the postmortem degradation and modification of muscle proteins during postmortem storage are the predominant factors influencing meat tenderness [9,10,11].

The study of protein changes permits the characterization of meat tenderization during postmortem storage as well as the monitoring of biomarkers characterizing tenderness [12,13].

In this context, we assume that knowledge of the postmortem degradation of the structural and cytoskeletal proteins of dromedary meat is necessary to better understand the postmortem processes as well as to develop strategies in carcass management aimed at producing meat with a competitive quality. Furthermore, a comprehensive understanding of the relationship between muscle protein changes during the first hours of postmortem and the slaughter age of dromedaries will help to design targeted interventions for obtaining better meat tenderness.

Thus, the objective of this study was to examine the effect of slaughter age and postmortem time on Sahraoui dromedary meat quality traits, with a particular focus on sarcoplasmic and myofibrillar protein changes.

## 2. Materials and Methods

### 2.1. Sample Collection

Six young (2 years ± 0.9 SD) and six adult (9 years ± 1.5 SD) male Sahraoui dromedaries reared in an extensive management system were randomly used for meat collection. Animals were exposed to the same preslaughter handling process. In the slaughterhouse, they were placed in lairage for about 12 h before being slaughtered, following the Algerian halal procedures, in a commercial slaughterhouse of the Ouargla region. After 3 h postmortem, *longissimus lumborum* (LL) muscles were removed from the last two lumbar vertebrae of each carcass and then transported to the biochemistry laboratory of University Kasdi Merbah Ouargla, Algeria. Muscles were kept at 12 °C until 12 h postmortem. Then, muscles were stored at 4 °C for 72 h, according to Al-Owaimer et al. [14]. Within 72 h, one slide was removed from each muscle at 6, 8, 10, 12, 24, 48 and 72 h. pH and water-holding capacity were determined at the University Kasdi Merbah Ouargla, Algeria. The determination of myofibrillar fragmentation index and protein changes was performed subsequently on samples stored at −20 °C and transported to the University of Foggia (Italy) in an insulated box filled with dry ice.

### 2.2. Meat Quality Properties

#### 2.2.1. pH Measurement

Muscle pH was determined according to Bendall [15]. Briefly, 1 g of ground muscle was homogenized with a polytron for 15 s, in 10 mL of buffer containing 5 mM sodium iodoacetate and 150 mM potassium chloride (pH 7). The pH was measured on the homogenate by using a pH meter equipped with combined glass electrode type HI9812-5 Hanna instruments (Hanna Instruments, Woonsocket, RI, USA).

#### 2.2.2. Water-Holding Capacity Determination

Water-holding capacity (WHC) was assessed according to the Grau–Hamm method [16]. A meat sample of 300 ± 5 mg was placed onto a previously desiccated filter paper, Whatmann No. 1 of 7 cm diameter; the paper with meat was then placed between two plexiglass plates. Loads of 2.25 kg were applied for 5 min. Circles of meat and released juice were then carefully drafted onto clear plastic sheets. The areas of meat spot and total liquid-infiltrated paper of each sheet of plastic were estimated using ImageJ software. The water-holding capacity was expressed as the total wet area less meat area (cm^2^) relative to the weight of the sample (g).

#### 2.2.3. Myofibrillar Fragmentation Index

The myofibrillar fragmentation index (MFI) was determined according to the protocol of Culler et al. [17], with some modifications. Briefly, 2 g of muscle was homogenized for 30 s in 20 mL of cold MFI buffer (100 mM KCl, 1 mM EGTA, 1 mM MgCl_2_ and 1 mM NaN_3_, pH 7). The homogenate was centrifuged at 1000× *g* for 15 min, after which the pellet was resuspended in 20 mL of MFI buffer, vortexed and centrifuged again. The sediment was resuspended in 10 mL of buffer and then filtered using a mesh screen to remove fat and connective tissue. The filtrate was used to quantify the protein concentration using the biuret method via a spectrophotometric assay. Subsequently, the concentration was adjusted to 0.5 mg/mL and the absorbance was read immediately at 540 nm. The MFI was calculated by multiplying the absorbance at 540 nm by 200.

### 2.3. Protein Analysis

#### 2.3.1. Protein Extraction

The isolation of sarcoplasmic and myofibrillar proteins was performed according to Marino et al. [13,18]. Briefly, 2.5 g of muscle was cut into small pieces (without fat or connective tissue) and homogenized using an Ultra Turrax at 10,000 rpm for 3 min with 10 mL of 0.03 M ice-cold phosphate buffer (pH 7) and a protease inhibitor cocktail (Sigma-Aldrich, St. Louis, MO, USA). Afterwards, the homogenate was centrifuged at 8000× *g* for 20 min at 4 °C. Supernatants (sarcoplasmic fraction) were collected and frozen at −80 °C. The resultant myofibrillar pellet was resuspended in 1 mL of denaturing buffer (8.3 M urea, 2 M thiourea, 64 mM dithiothreitol, 2% cholamidopropyl dimethyl hydroxypropane sulfonate, 2% Nonidet P-40, 10% glycerol and 20 mM Tris-HCl, pH 8), stirred overnight and centrifuged at 15,000× *g* for 20 min at 10 °C. After centrifugation, supernatants (myofibrillar fraction) were collected and stored at −80 °C. The protein concentration of each sarcoplasmic and myofibrillar extract (samples were assayed in triplicate) was determined by employing a 2-D Quant Kit (GE Healthcare), using serum albumin as a standard.

#### 2.3.2. SDS-PAGE Analysis

Myofibrillar and sarcoplasmic proteins were resolved according to the procedure of Marino et al. [13]. Briefly, an 8–18% sodium dodecyl sulfate-polyacrylamide gel electrophoresis (SDS-PAGE) gradient gel in a continuous buffer system was run at 24 mA/gel using a PROTEAN II xi system (Bio-Rad Laboratories, Hercules, CA, USA). Gels were stained with Coomassie Blue G250; images were acquired by the Chemi Doc EQ system (Bio-Rad Laboratories, Hercules, CA, USA) using a white-light conversion screen, after which they were analyzed with Image Lab software (version 5.2.1, Bio-Rad Laboratories, Hercules, CA, USA). Identification of the proteins was conducted via comparison with the Precision Plus Protein Standard—Broad Range (Bio-Rad Laboratories, Hercules, CA, USA), as well as with our previous identification conducted in the same conditions on myofibrillar and sarcoplasmic fractions of bovine *longissimus lumborum* [13,18]. The relative quantity of each band was determined as the percentage of the signal intensity of the defined bands in a lane.

#### 2.3.3. Western Blot

Western blot of troponin-T (TNNT) was performed as described by [7]. Briefly, myofibrillar proteins were separated on 12% acrylamide gels using Mini-PROTEAN Tetra cell (Bio-Rad Laboratories, Hercules, CA, USA). After a run, proteins were transferred onto nitrocellulose membranes (mini format, 0.2 μm nitrocellulose, Bio-Rad Laboratories) by a semidry transfer method (Trans-Blot Turbo Transfer System, Bio-Rad Laboratories) for 5 min at 25 V/2.5 A. All membranes were blocked for 1 h in a Tris-buffered saline solution containing 0.05% Tween-20 (TBS-Tween), including 5% of BSA as a blocking agent. Membranes were incubated with the primary antibodies: monoclonal anti-troponin-T produced in mouse (JLT-12; Sigma-Aldrich, St Louis, MO, USA; diluted 1:40,000). Then, membranes were washed, after which they were incubated for 1 h at room temperature with the secondary antibodies, goat anti-mouse-HRP (No 2554; Sigma-Aldrich, St Louis, MO; diluted 1:30,000). Blots were detected using the Clarity Western ECL Substrate (Bio-Rad Laboratories, Hercules, CA, USA). Images were acquired by the Chemi Doc EQ system and analyzed with Image Lab software (version 5.2.1, Bio-Rad Laboratories, Hercules, CA, USA) to determine the signal intensity of the intact and fragmented protein bands.

### 2.4. Statistical Analysis

All data were subjected to analysis of variance (ANOVA) using the GLM procedure of the SAS statistical software [19]. The mathematical model included the fixed effect due to slaughter age, postmortem time, slaughter age × postmortem time and random residual error. Results are presented as the least-squares means of the data for each age group, and the variability of the data is expressed by the standard error of the mean (SEM). All effects were tested for statistical significance (*p* < 0.05), and significant effects were reported. When significant effects were found (*p* < 0.05), Fisher’s LSD test was used to locate significant differences between means.

Principal component analysis (PCA) was applied to a matrix of 18 variables (pH, MFI, WHC, MYH, ACTN, DES, ACTA, TNNT, TPM, 30 kDa, MLY1, TNNI, sMYL2, fMYL2, PGM, ENO, CK and MB) using the PRINCOMP procedure of SAS to study the main tendencies in variation between the meat quality characteristics of adult and young dromedaries during postmortem time. The most significant 2 principal components were analyzed using factorial analysis.

## 3. Results and Discussion

### 3.1. Meat Quality Traits

The effect of slaughter age and postmortem time on the pH decline, MFI and WHC of the *longissimus lumborum* muscles from Algerian Sahraoui dromedaries are reported in Table 1. Slaughter age had a significant (*p* < 0.001) effect on the drop of muscle pH. In particular, compared to adult dromedaries, the *longissimus lumborum* muscles of young animals exhibited higher pH values at 8 and 72 h postmortem. Accordingly, Kadim et al. [5] found significantly higher pH values in younger animals (5.91 vs. 5.71 in young and adult camels, respectively) as a consequence of fiber types and muscle glycogen. Indeed, the proportion of red muscle fibers, with high glycogen content, increases with animal age and influences muscle metabolism and pH. Additionally, in cattle and goats it is reported that younger animals produce meat with a higher pH than older animals due to lower levels of glycogen [20,21].

During postmortem time, a progressive decrease in pH (*p* < 0.001) was observed, although with different rates in adult and young dromedaries. In particular, the LL muscles from adult dromedaries exhibited a more rapid pH decline starting from 8 h postmortem, while in young dromedaries a significant decrease was observed after 12 h postmortem. The faster pH decline of the LL muscles from adult dromedaries could be attributed to muscle glycogen reserves at the time of slaughter that accelerate glycolysis and the initial pH decline rate.

The myofibrillar fragmentation index (MFI) was significantly affected by both slaughter age (*p* ˂ 0.001) and postmortem time (*p* ˂ 0.05). It is noteworthy that the MFI is the most important index with which to measure the enhancement of meat tenderness and proteolysis, indicating both the I-band breaks and the loss of myofibril integrity. In this study, meat from young dromedaries showed higher MFI values than meat from adults. It is well-known that meat tenderization and the fragmentation of myofibrils are mainly related to muscle pH; overall, muscles with a high pH may tenderize more rapidly than muscles with lower pH values [22]. The higher fragmentation of myofibrils found in meat from young dromedaries could be related to the higher pH values found in the first hours postmortem, which led to the increased activity of endogenous enzymes on myofibrils [23].

As expected, postmortem time significantly affected myofibril fragmentation (*p* < 0.05), although this parameter increased at different rates in adult and young dromedaries. Meat from young dromedaries displayed an increase in the MFI after 12 h postmortem and remained constant thereafter; in meat from adult dromedaries an increase in the MFI was found only after 48 h postmortem. The more rapid myofibril fragmentation found in meat from young dromedaries starting from 12 h postmortem provides evidence of the faster resolution of rigor mortis and the early activation of endogenous enzymes in degrading myofibrils, thus highlighting differences in the proteolytic potential between young and adult dromedaries.

The water-holding capacity (WHC) is an essential measurement to estimate and assess juiciness, as well as, consequently, determining the appearance and palatability of the final product [24]. Slaughtering age affected the WHC values (*p* < 0.01); in particular, meat from adult dromedaries showed the lowest values of the WHC after 6 h of postmortem storage. It is well-known that a faster pH decline during the rigor phase affects the water-holding capacity in terms of lower drip loss [25]. In our study, the lowest WHC values of the LL muscles from adult dromedaries in the rigor phase could be due to the rapid pH fall and denaturation of muscle proteins, as previously suggested by Kadim et al. [5].

Postmortem time significantly affected the WHC (*p* ˂ 0.001) of dromedary meat. In particular, the ability of meat to retain moisture decreased during the first period of postmortem storage. A progressive increase in the WHC, until 24 h postmortem, was found for both adult and young dromedaries (the juice loss was about 120.47% and 223.81% in meat from young and adult dromedaries, respectively).

However, from 24 h up to 72 h of postmortem storage the WHC remained constant, suggesting that this phenomenon could be related principally to muscle acidification. Indeed, the pH fall and the rigor set induce shrinkage of the myofibril; thus, the space available for water within the myofibril is reduced, causing an increase in water loss.

### 3.2. SDS-PAGE of the Sarcoplasmic Fraction

The SDS-PAGE and the densitometric profile of sarcoplasmic proteins from the *longissimus lumborum* muscles of young and adult Sahraoui dromedaries during postmortem time is shown in Figure 1. The results show that the sarcoplasmic protein profile was significantly affected by slaughter age and postmortem time. The densitometric profiles evidenced that, at 6 h postmortem, both adult and young dromedaries were characterized by a similar profile in terms of the number and intensity of bands, while after 72 h postmortem meat from adult dromedaries displayed fewer sarcoplasmic protein bands compared to young dromedaries (27 and 19 bands in meat from young and adult dromedaries, respectively).

Image analysis results of the main sarcoplasmic proteins extracted from the *longissimus lumborum* muscles, as affected by slaughter age and postmortem time, are presented in Figure 2. Age at slaughter affected the electrophoretic profile of sarcoplasmic proteins, with adult dromedaries being characterized by higher values of myoglobin (*p* < 0.001) during all postmortem times. Myoglobin levels may reflect differences in the contractile and metabolic properties of muscle fibers. In our study the higher abundance of myoglobin found in the meat of adult dromedaries highlights the reddest meat, and confirms that the increase in myoglobin content is a process closely connected with an animal’s age, as previously reported [5,26]. In addition, meat from adult dromedaries showed higher values of phosphoglucomutase (PGM; *p* < 0.01) at 6 and 8 h postmortem, while meat from young dromedaries displayed a higher intensity of enolase (ENO; *p* < 0.001) and creatine kinase (CK; *p* < 0.01) after 8 and 24 h of postmortem storage, respectively.

Postmortem time affected the relative intensity of the sarcoplasmic proteins, mainly phosphoglucomutase (*p* < 0.01), enolase (*p* < 0.01) and creatine kinase (*p* < 0.01), both in adult and young dromedaries. Phosphoglucomutase (PGM) showed a decrease in intensity after 8 h postmortem in young dromedaries, while in adult dromedaries a decrease was found starting from 12 h postmortem. Phosphoglucomutase is an enzyme of the glycogen metabolism: it catalyzes the isomerization of glucose 1-phosphate, released from glycogen, to glucose 6-phosphate, which can then enter glycolysis. It has been shown that a greater PGM activity is usually related to a faster pH decline during the postrigor phase [27,28]. In our study, the faster decrease in PGM found in meat from young dromedaries, together with the pH results, suggest the involvement of PGM in the postmortem tenderization processes in such meat, in accordance with reports from other species [29,30].

Enolase (ENO) showed a significant increase, starting from 8 h and 10 h postmortem in young and adult dromedaries, respectively. Enolase is a key glycolytic enzyme catalyzing the conversion of 2-phosphoglycerate into phosphoenolpyruvate. Previous findings have reported that the abundance of enolase increases during postmortem storage [31,32] due to either modification, protein expression or minor degradation, highlighting the importance of glycolysis in the postmortem tenderization of meat. Consistently, in our study, the increase in enolase during the postmortem storage of LL highlights the association between this glycolytic enzyme and postmortem processes in dromedary meat.

The postmortem time also affected the relative intensity of creatine kinase (CK), showing a decrease (*p* < 0.01) starting from 12 h in meat from adult dromedaries. A different trend was observed for meat from young dromedaries, showing the lowest values at 10 h. Creatine kinase plays a critical role in maintaining ATP levels during the immediate postmortem period: it catalyzes reversibly the transfer of phosphate from creatine-phosphate to ADP, generating ATP and creatine [33].

### 3.3. SDS-PAGE of the Myofibrillar Fraction

The SDS-PAGE and the densitometric profile of myofibrillar proteins from the *longissimus lumborum* muscles of young and adult Sahraoui dromedaries according to post-mortem time is shown in Figure 3. The electrophoretic profile of myofibrillar proteins was affected by slaughter age and postmortem time, especially in terms of the intensity of protein bands. Indeed, although during postmortem time the electrophoretic profile of the myofibrillar fraction of both young and adult dromedaries was characterized by the same numbers of protein bands (23 and 26 at 6 h and 72 h postmortem), after 72 h postmortem young dromedaries showed a major intensity of polypeptides in the 30 kDa area.

Image analysis results of the main myofibrillar proteins extracted from the longissimus lumborum muscles, as affected by slaughter age and postmortem time, are presented in Figure 4. The slaughter age affected the percentage of desmin (DES, *p* < 0.01), myosin light chain 1 (MYL1, *p* < 0.01) and slow myosin light chain 2 (sMYL2, *p* < 0.01).

In particular, meat from adult dromedaries showed the lowest percentage of desmin during the prerigor and rigor phases (at 6, 8, 10, 12 and 24 h postmortem, respectively). In addition, meat from adult dromedaries was characterized by a higher percentage of MYL1 at 6, 8 and 10 h postmortem compared to young dromedaries. Concerning sMYL2, in the first hours postmortem meat from adult dromedaries showed the highest percentage of this isoform, while after 10 and 12 h postmortem was characterized by the lowest content than meat from young dromedaries.

A significant effect of postmortem time was found for myosin heavy chain (MHY; *p* < 0.01), alpha-actinin (ACTN; *p* < 0.05), desmin (DES; *p* < 0.01), troponin T (TNNT; *p* < 0.01), 30 kDa fragment (*p* < 0.01), myosin light chain 1 (MYL1; *p* < 0.01), slow myosin light chain 2 (sMYL2; *p* < 0.001) and fast myosin light chain 2 (fMYL2; *p* < 0.05).

Myosin is the most abundant myofibrillar protein that contributes to the structure and tensile strength of meat. The lowest values of this protein were found in the first postmortem phase (6 and 8 h) and in the postrigor phase (48 and 72 h). During postmortem time, protease activities are modulated by many factors, such as pH decline and accessibility to structural proteins as well as their modifications [10,34]. It has been reported that even a small change in the extent of actomyosin complex formation or the strength of the rigor bond could affect protease access to the substrate [34,35]. We retain that the changes found in the electrophoretic profile of MYH in dromedary meat reflect the post-mortem processes, and confirm that the extractability of proteins depends on the destructuration of the sarcomere by endogenous enzymes, as also reported in a previous study on bovine meat [13].

The changes in α-actinin abundance during postmortem time was observed only in meat from young dromedaries, showing the lowest values in the postrigor phase starting from 24 h postmortem. This finding confirms the role of α-actinin in maintaining the integrity of cells, since it is a major constituent of the Z-disk. It has also been reported that Z-lines are the first loci of postmortem structural alterations [33]. Therefore, any α-actinin degradation may lead to myofibrillar disorganization.

A decrease in desmin was found in the postrigor phase, at 48 and 72 h postmortem, only in meat from young dromedaries. Desmin is an intermediate filament and has a role in maintaining the integrity of muscle cells by connecting adjacent myofibrils at the level of their Z-lines, and the myofibrils to other cellular structures [36]. In a previous study on pork meat, early desmin proteolysis, starting from 24 h postmortem, was associated with the activation of calpain-1 [37]. Several studies on bovine meat [10,29] also reported that desmin degradation is an indicator of myofibrillar breakdown. In our study, the greater degradation of desmin together with the major increase in the MFI observed in meat from young dromedaries confirm the weakening of the myofibrillar structure due to the greater activity of calpain proteases and highlight the major proteolytic potential of meat from younger animals.

During postmortem time, myosin light chain 1 (MYL1) showed a different trend in meat from young and adult dromedaries. In particular, a decrease in the MYL1 percentage was observed for adults; conversely, an increase was found after 10 h postmortem in young muscle.

The differences in the MYL1 amount between adult and young dromedaries could be a reflection of the integrity of the actomyosin cross-bridges. Anderson et al. [29] reported that the rapid release of MYL1 from the myofibril is due to calpain-1 action, and that it is an indicator of early postmortem proteolysis as well as, potentially, tenderization. In our study, the lowest value of MYL1 found in the meat of young dromedaries during first-phase postmortem could be due to the greater release of this myofibrillar protein into a soluble fraction, confirming that MYL1 could also be an indicator of myofibril destructuration in dromedary meat, as reported previously on bovine meat [18].

### 3.4. Western Blot of TnT

It has been largely reported that troponin T is one of the main proteins used as a marker for the ongoing proteolysis and tenderness prediction [10,34]. Therefore, a Western blot of troponin T was performed to better explain the changes detected between meat from adult and young dromedaries during postmortem storage, as well as in an effort to confirm the SDS-PAGE results.

A representative Western blot of troponin T from the *longissimus lumborum* muscles of dromedaries as affected by slaughter age and postmortem time is reported in Figure 5, while the image analysis results are presented in Table 2.

A Western blot of troponin T (TNNT) showed the presence of five immunoreactive bands: 38, 36 and 34 kDa represent isoforms of intact TNNT, while the 30 and 28 kDa bands were degraded products of TNNT.

Referring to intact isoforms of TNNT, image analysis results revealed that both slaughter age and postmortem time significantly affected the relative intensity of 38, 36 and 34 kDa protein bands. In particular, the 38 kDa and 34 kDa intact isoforms showed higher values in meat from young dromedaries at 6 and 8 h postmortem (*p* ˂ 0.05), while meat from adult dromedaries showed the highest intensity only of the 36 kDa band (*p* < 0.01) until 48 h postmortem.

As expected, post-mortem time significantly affected the degradation of intact TNNT isoforms (38 kDa, 36 kDa and 34 kDa, respectively), but with different rates. Referring to the 38 kDa protein, in adult dromedaries a decrease (*p* < 0.01) was found starting from 10 h postmortem, while in young dromedaries a progressive degradation was observed during postmortem time. An effect of postmortem time was found for the 36 kDa band in adult dromedaries, with a significant decrease (*p* < 0.05) after 10 h postmortem. While referring to the 34 kDa band, a progressive decrease was found in young dromedaries, showing the lowest values starting from 24 h postmortem.

However, these results highlight that meat from young dromedaries was characterized by the highest percentage of intact TNNT isoforms (sum of all the isoforms) in the first phase of postmortem time, while the end of treatment displayed similar values to meat from older animals. As a consequence, the percentage of intact TNNT reduction during postmortem time was about 52.86% and 46.87% in meat from young and adult dromedaries, respectively.

The major degradation of intact TNNT isoforms in meat from young dromedaries during postmortem time could be due to the fiber characteristics and the greater activity of endogenous enzymes in degrading myofibrils, as also supported by the MFI results. Consistent with our data, Cruzen et al. [38] observed less troponin-T degradation in mature beef muscles compared to young ones, and suggested a reduced proteolysis potential in mature cattle compared to calves due to a lower ratio of μ-calpain/total calpastatin activity.

It is known that 30 and 28 kDa fragments are the two major polypeptides generated from troponin-T degradation [34]. Concerning the degraded isoforms, the 30 kDa band was affected by slaughter age (*p* < 0.05) and postmortem time (*p* < 0.001). Meat from adult dromedaries was characterized by the highest percentage at 6, 8 and 12 h postmortem, while after 24 h meat from young dromedaries showed higher values.

During postmortem time, a progressive increase in the 30 and 28 kDa bands was found in both young and adult dromedaries, with different rates. Meat from adult dromedaries showed an increase in the 30 kDa fragment starting from 10 h postmortem, while in meat from young dromedaries a progressive increase was found. Referring to the 28 kDa polypeptide, in adult dromedaries this polypeptide appeared after 48 h postmortem, while in young dromedaries it appeared earlier, after 10 h postmortem.

The greater amount of TNNT fragments found in meat from young animals during the postmortem time confirms the more rapid and extensive degradation of intact TNNT observed, highlighting the more intense degradation of myofibrils and proteolysis in younger animals. Therefore, our results revealed differences in the meat proteolytic potential due to the slaughter age of camels, suggesting that knowledge of the factors that influence muscle proteome and its evolution during postmortem changes is necessary to set targeted interventions to improve the aging process of dromedary meat cuts.

### 3.5. Relationship between Sarcoplasmic and Myofibrillar Proteins Patterns and Dromedary Meat Quality Characteristics

The principal component analysis of the pH, MFI, WHC and relative intensity of the myofibrillar and sarcoplasmic proteins of the *longissimus lumborum* muscles from young and adult dromedaries during postmortem time (6, 8, 10, 12, 24, 48 and 72 h) is reported in Figure 6.

The PCA applied to the mentioned variables accounted for 57.3% of the total variance, with 36.51% explained by PC1 and 20.79% explained by PC2. Phosphoglucomutase, troponin T and the 30 kDa fragment were the main factors explained by PC1, the latter being negatively related to the principal component. Tropomyosin, desmin, myoglobin and α-actinin were dominating factors along with PC2, with myoglobin and α-actinin negatively related to the principal component. Apart from postmortem time, the PCA score plot evidenced a clear separation of meat from adult and young dromedaries along with the second principal component, with meat from younger animals being characterized by a higher pH, together with higher contents of TPM, DES and CK.

Furthermore, the score plot evidenced that samples moved along the first principal component as the postmortem time advanced, with 6, 8, 10 and 12 h postmortem being located on the right side of the axis, while meat samples starting from 24 h postmortem displayed negative scores.

The PCA analysis also showed that meat from young dromedaries starting 24 h postmortem was located in a zone of the plot characterized by higher levels of the myofibrillar fragmentation index, 30 kDa polypeptide and enolase, overall confirming the greater rupture of myofibrils and proteolysis in younger animals.

## 4. Conclusions

Data from the current study highlight that both slaughter age and postmortem time may affect the properties of the *Longissimus lumborum* muscles of Sahraoui dromedaries and provide insights into protein changes during the conversion of muscle into meat. Postmortem time in adult and young dromedaries resulted in different meat tenderization rates. Young dromedaries evidenced a slower acidification process with greater myofibril fragmentation throughout the post-mortem time as a result of the endogenous enzyme activity.

In particular, an early and intense postmortem proteolysis involves meat from young dromedaries, as evidenced by the degradation of key proteins of the tenderization process, such as desmin and α-actinin. These findings have also been supported by the more intense and faster troponin T degradation found in younger animals, which confirmed a different kinetic in the disruption and degradation of myofibrils during the postmortem time, highlighting a different proteolytic potential between dromedaries of different ages.

## Figures and Tables

**Figure 1 foods-11-00732-f001:**
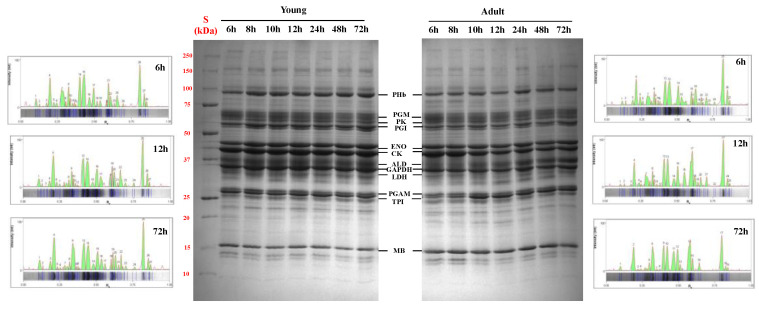
Representative SDS-PAGE (8–18%) and densitometric profiles of sarcoplasmic proteins from the *longissimus lumborum* muscles of young and adult dromedaries after 6, 8, 10, 12, 24, 48 and 72 h postmortem (St = standard; PHb = phosphorylase b; PGM = phosphoglucomutase; PK = pyruvate kinase; PGI = phosphoglucose isomerase; ENO = enolase; CK = creatine kinase; ALD = aldolase; GAPDH = glyceraldehyde phosphate dehydrogenase; LDH = lactate dehydrogenase; PGAM = phosphoglycerate mutase; TPI = triosephosphate isomerase; and MB = myoglobin).

**Figure 2 foods-11-00732-f002:**
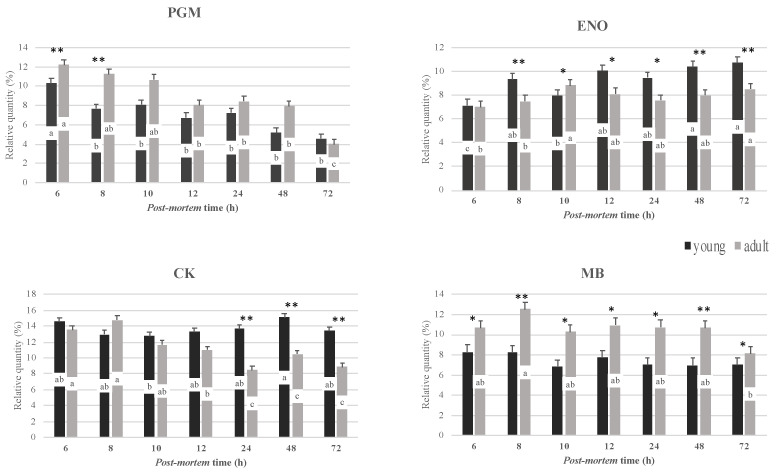
Percentages (%) of the main sarcoplasmic proteins from the *longissimus lumborum* muscles of young and adult dromedaries after 6, 8, 10, 12, 24, 48 and 72 h postmortem (PGM = phosphoglucomutase; ENO = enolase; CK = creatine kinase; MB = myoglobin; ∗ = *p* < 0.05; ∗∗ = *p* < 0.01 slaughter age effect; a–c = *p* < 0.05 postmortem time effect; and means ± SEM).

**Figure 3 foods-11-00732-f003:**
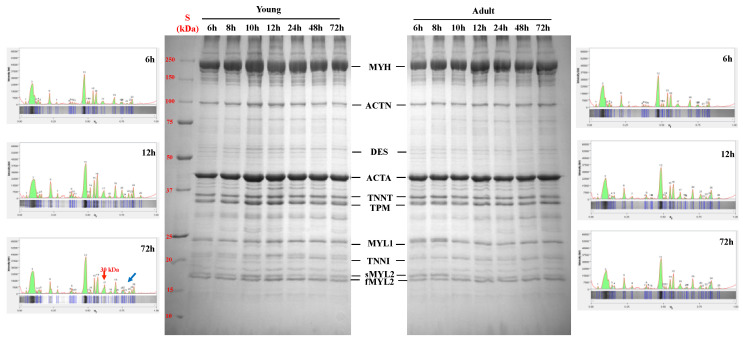
Representative SDS-PAGE (8–18%) and densitometric profile of myofibrillar proteins from the *longissimus lumborum* muscles of young and adult dromedaries after 6, 8, 10, 12, 24, 48 and 72 h postmortem (St = standard; MYH = myosin heavy chain; ACTN = α-actinin; DES = desmin; TNNT = troponin-T; TPM = tropomyosin; MYL1 = myosin light chain 1; TNNI = troponin-I; sMYL2 = slow myosin light chain 2; and fMYL2 = fast myosin light chain 2).

**Figure 4 foods-11-00732-f004:**
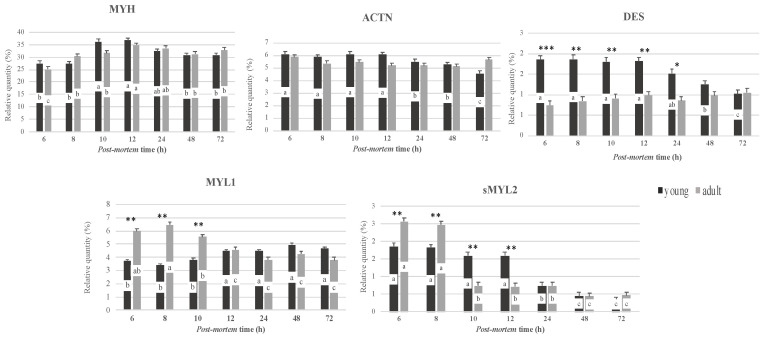
Percentages (%) of the main myofibrillar proteins from the *longissimus lumborum* muscles of young and adult dromedaries after 6, 8, 10, 12, 24, 48 and 72 h postmortem (MYH = myosin heavy chain; ACTN = α-actinin; DES = desmin; MYL1 = myosin light chain 1; sMYL2 = slow myosin light chain 2; ∗ = *p* < 0.05; ∗∗ = *p* < 0.01; ∗∗∗ = *p* < 0.001 slaughter age effect; a–c = *p* < 0.05, postmortem time effect; and means ± SEM).

**Figure 5 foods-11-00732-f005:**
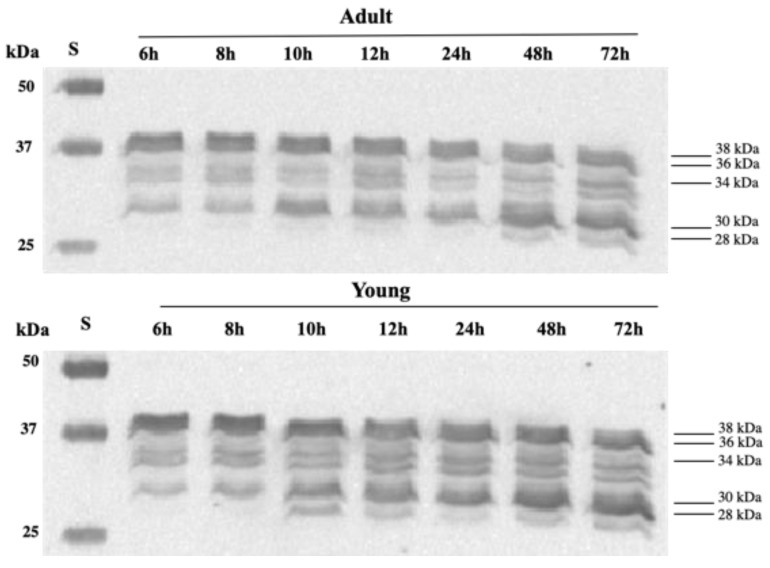
Representative Western blot of troponin-T of the *longissimus lumborum* muscles of young and adult dromedaries after 6, 8 10, 12, 24, 48 and 72 h postmortem (troponin-T isoforms: 38, 36 and 34 kDa (intact forms); 30 and 28 kDa (degraded forms)).

**Figure 6 foods-11-00732-f006:**
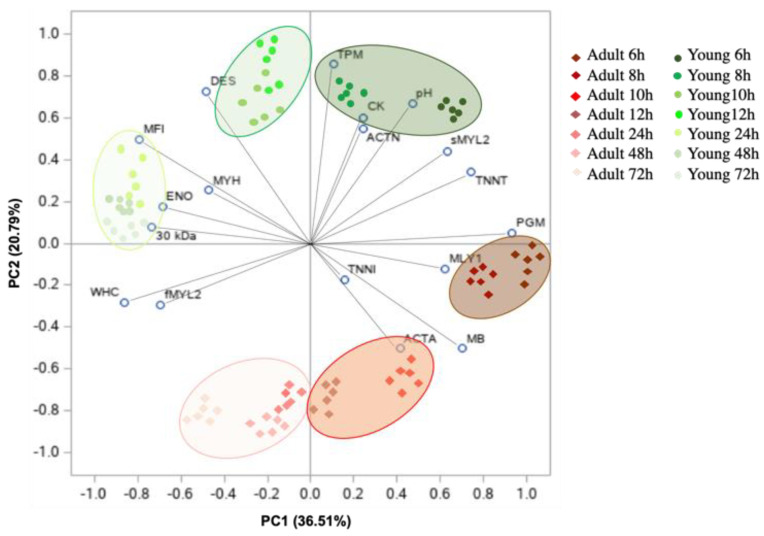
Principal component analysis (PCA) of the pH, MFI, WHC and relative intensity of the myofibrillar and sarcoplasmic proteins of the *longissimus lumborum* muscles from young and adult dromedaries during postmortem time (6, 8, 10, 12, 24, 48 and 72 h).

**Table 1 foods-11-00732-t001:** pH decline, myofibrillar fragmentation index (MFI) and water-holding capacity (WHC) of the *longissimus lumborum* (LL) muscles of Sahraoui dromedaries after 6, 8, 10, 12, 24, 48 and 72 h (h) postmortem (means ± SEM), according to slaughter age.

		Postmortem Time (h)							Effects, *p*			
		6	8	10	12	24	48	72	SEM	Age	Time	Age × Time
pH	Young	6.43 a	6.34 Aab	6.22 ab	6.20 b	5.99 c	5.98 c	5.94 Ac	0.07	***	***	*
	Adult	6.32 a	5.99 Bb	6.00 b	5.98 b	5.87 bc	5.89 bc	5.67 Bc				
MFI	Young	77.89 Ab	78.91 Ab	92.72 Aab	99.75 Aa	101.45 Aa	98.12 Aa	97.48 Aa	5.61	***	*	NS
	Adult	55.85 Bb	56.40 Bb	57.17 Bb	60.09 Bb	70.28 Bab	77.43 Ba	78.41 Ba				
WHC (cm^2^/g)	Young	12.21 Ac	15.61 bc	19.42 b	17.02 bc	26.92 a	24.01 a	25.42 a	1.25	**	***	*
	Adult	7.98 Bc	14.85 b	16.64 b	15.90 b	25.84 a	24.88 a	25.50 a				

NS = not significant; * = *p* < 0.05; ** = *p* < 0.01; and *** = *p* < 0.001. A, B = *p <* 0.05 in the column (slaughter age effect); a–c = *p* < 0.05 in the row (postmortem time effect).

**Table 2 foods-11-00732-t002:** Image analysis results (%) of troponin T Western blot of the *longissimus lumborum* muscles of young and adult dromedaries after 6, 8 10, 12, 24, 48 and 72 h postmortem (troponin-T isoforms: 38, 36 and 34 kDa (intact forms); 30 and 28 kDa (degraded forms)).

		Postmortem Time (h)								Effects, *p*		
		6	8	10	12	24	48	72	SEM	Age	Time	Age × Time
Intact forms												
38 kDa	adult	71.58 Ba	68.41 Ba	59.22 b	56.33 b	53.53 b	40.21 c	36.81 c	1.92	*	**	*
	young	84.12 Aa	77.41 Ab	58.20 c	58.55 c	50.33 d	40.77 e	37.36 e				
36 kDa	adult	10.80 Aa	10.73 Aa	6.35 Ab	6.20 Ab	6.99 Ab	6.67 Ab	5.25 b	0.61	**	*	*
	young	4.67 B	4.10 B	4.51 B	4.04 B	4.33 B	4.30 B	4.10				
34 kDa	adult	2.20 B	2.15 B	2.31	2.34	2.86	2.87	2.88	0.43	*	*	*
	young	3.72 Aa	3.95 Aa	2.72 ab	2.99 ab	2.51 b	2.40 b	2.15 b				
Degraded forms												
30 kDa	adult	15.42 Ad	18.71 Ad	32.12 c	35.13 Abc	36.62 Bb	48.91 a	52.70 a	1.38	*	***	*
	young	7.49 Bf	14.54 Be	29.64 d	30.76 Bd	41.04 Ac	50.86 b	54.35 a				
28 kDa	adult	ND	ND	ND	ND	ND	1.34 b	2.36 a	0.25	*	**	*
	young	ND	ND	4.94 a	3.65 b	1.78 c	1.78 c	2.04 c				

ND = not detected; NS = not significant; * = *p* < 0.05; ** = *p* < 0.01; and *** = *p* < 0.001. A, B = *p* < 0.05 in the column (slaughter age effect); a–f = *p* < 0.05 in the row (postmortem time effect).

## Data Availability

Not applicable.

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
