# Peer review of "Postmortem Muscle Protein Changes as a Tool for Monitoring Sahraoui Dromedary Meat Quality Characteristics"

_foods, 2022, doi:10.3390/foods11050732_

Round 1

Reviewer 1 Report

General comments:

This study is to investigate the effects of slaughter age (2 vs 9 years) and post-mortem time (6, 8, 10, 12, 24, 48 and 72h) on meat quality and protein changes of longissimus lumborum muscles of Algerian Sahraoui dromedary. There are several major concerns. The first one is lack of meat tenderness measurement (shear force or textual properties) since authors outline the importance for dromedary in introduction part (Line 44, line 50). The second is the WHC measurement which is not solid for readers. The most concern is the many proteins identification and quantification only derived from the SDS-PAGE and optical density analysis, which not confirmed by other techniques such as MS. Any comparison and PCA analysis based on that data are insignificant, resulting in labile conclusion. 

Minor suggestions:

Line 56-58 Description of proteomics techniques is not related to this study and need to be omitted.

Line 90 Add company information of Hanna HI9812-5 instrument

Line 93 “300 ± 5 mg”, The sentence cannot begin with number. In addition, any reference of WHC measurement since it seems not in ordinary way by assessing the water loss by treatments.

Line 104 MgCl2 needs subscript

Line 105 pallet changes to pellet

Line 130 what is continuous buffer system

Line 231-234 combined to one paragraph

Line 243 how to identify myoglobin band in the SDS-PAGE ? also for other proteins

Author Response

Reviewer 1

General comments:

This study is to investigate the effects of slaughter age (2 vs 9 years) and post-mortem time (6, 8, 10, 12, 24, 48 and 72h) on meat quality and protein changes of longissimus lumborum muscles of Algerian Sahraoui dromedary. There are several major concerns. The first one is lack of meat tenderness measurement (shear force or textual properties) since authors outline the importance for dromedary in introduction part (Line 44, line 50). The second is the WHC measurement which is not solid for readers. The most concern is the many proteins identification and quantification only derived from the SDS-PAGE and optical density analysis, which not confirmed by other techniques such as MS. Any comparison and PCA analysis based on that data are insignificant, resulting in labile conclusion. 

Au: As the reviewer noted the instrumental tenderness measurement are not reported in the present study however, the principal aim was to investigate the key proteins related to the tenderization process and their changes during the first hours’ post-mortem (e.g. troponin T). The paper put emphasis on the study of the protein profile which is a more sensitive approach in relation to the traditional production protocols of dromedary meat that provide for limited aging time. In addition, the WHC description was improved. We understand, also, the concern of the reviewer about the identification and quantification of the proteins, but western blot was performed to troponin t identification. In addition, the identification of many proteins was done by comparison with molecular weight standards and also according to our previous experience in SDS-PAGE profile and MS identifications. (Marino et al. (2014). Meat Sci., 98(2), 178–186. https://doi.org/10.1016/j.meatsci.2014.05.024; Marino et al. (2015). J. Anim. Sci., 93(3), 1376–1387. https://doi.org/10.2527/jas.2014-835; della Malva et al. (2017). Meat Sci., 13, 74-81.https://doi.org/10.1016/j.meatsci.2017.04.235; della Malva et al. (2022). Meat Sci., 184, 108686.https://doi.org/10.1016/j.meatsci.2021.108686)

Minor suggestions:

Line 56-58 Description of proteomics techniques is not related to this study and need to be omitted.

Au: According to reviewer suggestion, the text has been improved.

Line 90 Add company information of Hanna HI9812-5 instrument

Au: It has been added.

Line 93 “300 ± 5 mg”, The sentence cannot begin with number. In addition, any reference of WHC measurement since it seems not in ordinary way by assessing the water loss by treatments.

Au: The sencence has been modified, accordingly. However, reference of WHC measurement has been added.

Line 104 MgCl2 needs subscript

Au: It has been corrected.

Line 105 pallet changes to pellet

Au: It has been changed.

Line 130 what is continuous buffer system

Au: The continuous buffer is a system that use the same buffer (at constant pH) in the gel samples and both electrode reservoirs.

Line 231-234 combined to one paragraph

Au: It has been corrected.

Line 243 how to identify myoglobin band in the SDS-PAGE ? also for other proteins

Au: The manuscript has been improved to clarify the proteins identification.

New and added parts in the manuscript are highlighted in red.

Reviewer 2 Report

The manuscript entitled “Post-mortem muscle protein changes as a tool for monitoring Sahraoui dromedary meat quality characteristics”, authored by Hanane Smili, et al., will provide information for the investigation of muscle proteome, aim to set targeted interventions to improve the tenderness of dromedary meat cuts.

The manuscript still needs partial technical improvements.

  1. Some sentences are grammatically incorrect or inappropriate.
    • Line 41, “draught” maybe change into other words.
    • Line 60, “post-mortem” should be italicized.
    • Line 211, “rigor development” may be more appropriate to be replaced by “rigor phase”.
    • Line 228, “P<0.05” should be italicized.
  2. Line 79, Does “At different post-mortem time (6, 8, 10, 12, 24, 48, 72 hours) one slide was removed from each muscle” mean that the muscle was kept in 4°C for 6, 8, 10 extra hours? Please add details of how the sample was taken.
  3. Line 185-186, About the reason of adult dromedaries faster pH decline should be more discussion.
  4. Line 197, “that led to increased activity of endogenous enzymes on myofibril” should be added documentary evidence.
  5. Line 198, the “As expected, post-mortem time significantly affected myofibrils fragmentation” as the conclusion of this paragraph, but this paragraph mainly discussed the difference between adult dromedaries and young dromedaries. This conclusion should be changed into another aspect.
  6. Lines 352-353 please add a discussion of the reasons for the differences between adult dromedaries and young dromedaries.

Author Response

Reviewer 2

The manuscript entitled “Post-mortem muscle protein changes as a tool for monitoring Sahraoui dromedary meat quality characteristics”, authored by Hanane Smili, et al., will provide information for the investigation of muscle proteome, aim to set targeted interventions to improve the tenderness of dromedary meat cuts.

Au: We would like to thank this reviewer for the comments that have contributed to improving our manuscript.

The manuscript still needs partial technical improvements.

  1. Some sentences are grammatically incorrect or inappropriate.
  • Line 41, “draught” maybe change into other words.

Au: It has been corrected.

  • Line 60, “post-mortem” should be italicized.

Au: It has been corrected.

  • Line 211, “rigor development” may be more appropriate to be replaced by “rigor phase”.

Au: It has been changed.

  • Line 228, “P<0.05” should be italicized.

Au: It has been corrected.

  1. Line 79, Does “At different post-mortem time (6, 8, 10, 12, 24, 48, 72 hours) one slide was removed from each muscle” mean that the muscle was kept in 4°C for 6, 8, 10 extra hours? Please add details of how the sample was taken.

Au: Details of procedure have been added in the revised manuscript.

  1. Line 185-186, About the reason of adult dromedaries faster pH decline should be more discussion.

Au: This issue was also discussed in lanes 174-180.

  1. Line 197, “that led to increased activity of endogenous enzymes on myofibril” should be added documentary evidence.

Au:  Reference has been added in the revised manuscript.

  1. Line 198, the “As expected, post-mortem time significantly affected myofibrils fragmentation” as the conclusion of this paragraph, but this paragraph mainly discussed the difference between adult dromedaries and young dromedaries. This conclusion should be changed into another aspect.

Au: The reviewer refers to a sentence which is not a conclusion but reports the effects of post-mortem time and slaughter age. In the subsequent lines (202-206), the differences between adult and young dromedaries is further discussed and explained.

  1. Lines 352-353 please add a discussion of the reasons for the differences between adult dromedaries and young dromedaries.

Au: We agree with reviewer, indeed in the lanes 355-358 we hypothesized the reason of the differences between adult and young dromedaries.

New and added parts in the manuscript are highlighted in red.

Reviewer 3 Report

Dear Authors,

I reviewed the manuscript  (ID: foods-1604305) entitled: “Post-mortem muscle protein changes as a tool for monitoring Sahraoui dromedary meat quality characteristics”. This manuscript presents the relationship between muscle protein changes during the first hours’ post-mortem and slaughter age of dromedaries - very interesting and important animals in Algeria.

A very detailed methodology allows other scientists to repeat or continue their research in this area. Correctly selected methodologies and tools allowed the authors to present the results in an interesting way and to discuss them.
I have found this paper of interest. However, please consider a few suggestions. Please, see details below:
S1. Please make a few editorial corrections:

correct in the whole manuscript

  • insert a space between the number and the unit, e.g. 72 h; 20 °C; p < 0.05
  • delete the space between the literature numbers, e.g. [1,2,3]

furthermore correct

  • L90: “type Hanna HI9812-5 instrument.” to correct “type HI9812-5 Hanna Instruments.”
  • L99: – cm2 – insert superscript
  • L104: MgCl2; 105: NaN3 -  insert subscript
  • L105: pH 7.0 - delete .0

S2. Point 2.2.2. Water holding capacity determination - please provide the name of the method and reference literature.

Author Response

Reviewer 3

Dear Authors,

I reviewed the manuscript (ID: foods-1604305) entitled: “Post-mortem muscle protein changes as a tool for monitoring Sahraoui dromedary meat quality characteristics”. This manuscript presents the relationship between muscle protein changes during the first hours’ post-mortem and slaughter age of dromedaries - very interesting and important animals in Algeria.

A very detailed methodology allows other scientists to repeat or continue their research in this area. Correctly selected methodologies and tools allowed the authors to present the results in an interesting way and to discuss them.
I have found this paper of interest. However, please consider a few suggestions. Please, see details below:
S1. Please make a few editorial corrections:

Au: We gratefully acknowledge the positive comments made by this review. The manuscript has been improved according to reviewer’s suggestions.

correct in the whole manuscript

  • insert a space between the number and the unit, e.g. 72 h; 20 °C; p < 0.05

Au: The text has been revised accordingly.

  • delete the space between the literature numbers, e.g. [1,2,3]

Au: The text has been revised accordingly.

furthermore correct:

  • L90: “type Hanna HI9812-5 instrument.” to correct “type HI9812-5 Hanna Instruments.”

Au: It has been corrected.

  • L99: – cm2 – insert superscript

Au: It has been corrected.

  • L104: MgCl2; 105: NaN3 -  insert subscript

Au: It has been corrected.

  • L105: pH 7.0 - delete .0

Au: It has been corrected.

S2. Point 2.2.2. Water holding capacity determination - please provide the name of the method and reference literature.

Au: The name of the method and reference of WHC determination have been added in the revised manuscript.

New and added parts in the manuscript are highlighted in red.

Round 2

Reviewer 1 Report

Though solid evidence have been given between the relationship between the proteolysis (or indicated by myofibrillar fragmantation index) and meat tenderness in many published research, the author cannot given a conclusion (Line 26-27) without measuring the meat tenderness. The major concern is also for the muscle proteome, which is only conducted by SDS-PAGE, despite the western-blot of troponin-T. The SDS-PAGE indeed indicates the protein bands changes during the postmortem aging and can compared with different treatments. However, the authors cannot label the one specific protein at protein band of SDS-PAGE image (Figure 1 and Figure 3) according to the molecular weight since one protein band can include many proteins of different isoforms, degraded fragments and post-translation modification. The optical density analysis of protein bands only indicated the proteins abundance at that range of molecular weight, which should not confirmed as specific protein like, myoglobin, PGM, ENO, CK and etc. Thus all descriptions and discussions concerning the specific protein by the estimation of band molecular weight are not valid and convinced for readers.

Author Response

AU: Thank you for your remark. We understand concerns about the instrumental tenderness measurement and we want to explain why this was not performed. Based on our experience in the meat tenderization rate, we decided to evaluate the myofibrillar fragmentation index as a method strictly connected with myofibrils destructuration by endogenous enzymes and consequently meat tenderness. However, some data about the sensorial evaluation of dromedary meat tenderness were presented at ICoMST 2019 congress (Smili et al., 2019. Sensory proprieties of Sahraoui dromedary meat and relationship with quality traits, myofibrillar and sarcoplasmic proteins. ICOMST 2019, Proceedings of the 65th International Congress of Meat Science and Technology – Potsdam (Germany), pp. 756-758) showing a significant effect of slaughter age on tenderness, juiciness and overall liking, particularly young dromedaries showed more tender and juicy meat than adult dromedaries.

About muscle proteome, we understand concerns that one protein band can include many proteins of different isoforms, degraded fragments and post-translation modification. In this paper, the identification of the proteins was done by comparison with previous identification conducted in the same condition on myofibrillar and sarcoplasmic fractions of bovine and horse longissimus lumborum (Marino et al. (2013). Meat Sci., 95, 281-287. http://dx.doi.org/10.1016/j.meatsci.2013.04.009; Marino et al. (2014). Meat Sci., 98(2), 178–186. https://doi.org/10.1016/j.meatsci.2014.05.024; Marino et al. (2015). J. Anim. Sci., 93(3), 1376–1387. https://doi.org/10.2527/jas.2014-835; della Malva et al. (2022). Meat Sci., 184, 108686. https://doi.org/10.1016/j.meatsci.2021.108686). The manuscript was revised to add this information.

However, we agree with the reviewer, about the need to add knowledge on the proteome analysis on dromedary meat by means of two-dimensional techniques and mass spectrometry to identify potential biomarkers to monitor post-mortem changes that we wish to conduct in further studies.